# The frequency of workplace violence against healthcare workers and affecting factors

**Hıdır Sari** [1]*, **İsmail Yildiz**[2], **Senem Çağla Baloğlu**[3], **Mehmet Özel**[4], **Ronay Tekalp**[5]

**1** Dicle University Faculty of Medicine, Department of Public Health, Diyarbakır, Turkey, **2** Dicle University Faculty of Medicine, Department of Biostatistics and Medical Informatics, Diyarbakır, Turkey, **3** Dicle University, Diyarbakır, Turkey, **4** Department of Emergency Medicine, Diyarbakır Gazi Yasargil Training and Research Hospital, University of Health Sciences, Diyarbakır, Turkey, **5** Dicle University, Office of Legal Affairs, Diyarbakır, Turkey

* hsaridr@yahoo.com

## Abstract

### Background

Workplace violence has become a global issue, especially among healthcare workers. This study aimed to determine the influencing factors and legal processes of workplace violence incidents, as well as the frequency of workplace violence in a tertiary hospital.

### Methods

This observational, descriptive, retrospective frequency study was conducted between January 2020 and March 2022. This study examined the workplace violence records of 135 healthcare professionals at a tertiary hospital's Patient Rights and Employee Safety and Law departments. Factors affecting workplace violence were categorized as noncompliance with the procedure, communication, and dissatisfaction.

### Results

Workplace violence frequency was observed in the cumulative total of 10821 healthcare workers at 1.2%. In terms of workplace violence types, 71.9% were verbal and 28.1% were physical. In terms of exposure to workplace violence, doctors accounted for 62.3%, nurses for 20%, and medical secretaries for 7.4%. Most cases were observed in outpatient clinics (34.8%), followed by emergency departments (25.9%). Among the main reasons for workplace violence against healthcare workers, non-compliance with procedures (49.6%), communication (27.4%), and dissatisfaction (23.1%) were identified. Legal aid was provided to all notifications of workplace violence. 37.1% were not prosecuted, 55.5% were under investigation, 4.4% were accepted indictments, and 3.0% were punished by a judicial fine.

### Conclusion

This study can provide significant contributions to the formulation of workplace violence prevention policies and programs by analyzing white-code notifications for workplace violence frequency and preventable factors. Healthcare workers may have underreported workplace

**Data Availability Statement:** Due to research ethics committee restrictions, we are unable to make the data publicly available due to restrictions in the approved protocol. Coded data may be requested from the corresponding author for

nonprofit and justified scientific reasons from academic institutions. For more information, please contact Dicle University Ethics Committee at kuruletikdiyar@gmail.com.

**Funding:** The authors received no specific funding for this work.

**Competing interests:** The authors have declared that no competing interests exist.

violence events due to the length of the proceedings and the perceived lack of protection from legal regulations.

## Introduction

A violent act is defined by the World Health Organization (WHO) as: "The intentional use of physical force or power against another individual, a group, or community, which causes an injury, death, psychological harm, maldevelopment or deprivation, or has a high likelihood of resulting in such injury, death, psychological harm, maldevelopment, or deprivation" [1]. WHO published the first comprehensive report on violence and health in 2002, examining violence as a global health issue. Violence is responsible for more than 1.6 million deaths and millions of injuries each year, resulting in physical, sexual, reproductive, and mental health problems [2].

Workplace violence (WPV) is an act or threat of violence that ranges from verbal abuse to physical assault at the workplace or against persons in charge, according to the National Institute for Occupational Safety and Health (NIOSH). Psychological problems and physical injuries are just a few of the consequences of WPV. Health and welfare workers are at the highest risk of being absent from work because of non-fatal violence [3]. In a report published by the International Labor Organization (ILO) in 1998 on occupational injury statistics, according to the definition of a work accident, was defined as "an unexpected and unplanned event, including non-consensual acts of violence that result in personal injury, illness, or death, arising from work or in connection with work", acts of violence are categorized as work accidents. The term "workplace violence" refers to acts of violence against employees, employers, and other individuals at work as well as outside the workplace. [4]. As a consequence of WPV, productivity drops, turnover increases, absenteeism increases, consulting costs increase, and staff morale decreases [5].

Healthcare workers (HCWs) are particularly affected by WPV, which has become a global social problem [6]. In European countries, 4% of active HCWs have reported that they had been subjected to verbal or physical WPV from patients or visitors [7]. According to a national survey on WPV in Turkey conducted by Pinar et al, 6.8% of health workers have been exposed to physical violence, 43.2% to verbal violence, and 44.7% to both forms [8]. In a systematic review and meta-analysis study conducted in China, 19.33% of WPV against HCWs were perpetrated by patients or visitors over a one-year period [9]. According to a study conducted in a tertiary care hospital in the USA, 34.4% of the health workers reported verbal or physical WPV, 31.9% both verbal and 13.5% physical assault. Among those who experienced physical or verbal WPV, 60.2% showed at least one post-traumatic symptom, 9.4% lost their jobs, and 30.1% considered quitting their careers [10]. In a study conducted in India; Even although WPV among HCWs was detected at a high prevalence (34.5%), it was reported at a low rate (23.5%), and HCWs were unaware of the reporting mechanism and regulations regarding workplace violence protection (24.6%) [11]. A study conducted in Bangladesh; the study revealed that violence against health workers is under-reported; 96% of WPV were cases of physical violence, and one-third of violence cases caused strikes and interruption of health services [12].

According to studies conducted across the globe, WPV events against HCWs are influenced by a variety of factors. Several of these factors have been prevented [13–15]. Several studies have been conducted worldwide to better understand the risk factors and predictors that cause

such WPV events and to assess strategies for reducing them. Several factors have been reported that contribute to violence against healthcare professionals, including the behavior of patients and employees, hospital environment, waiting times, and professional roles [16,17].

Guidelines and technical tools for assessing WPV have been developed in several countries. Italy's Ministry of Health issued recommendations in 2007 on managing violence among healthcare workers [18]. As part of its efforts to create a safe healthcare environment, the U.S. Occupational Safety and Health Administration (OSHA) released the "Guidelines for preventing WPV for health care and social service workers" in 2004 [19]. The white code application is a legal aid program offered by the Ministry of Health and the affiliated health institutions upon request of HCW in legal actions and proceedings undertaken within the scope of criminal law due to crimes committed against them during the delivery of health services by the Ministry of Health of the Republic of Turkey (MHRT) according to legal regulations that entered into force in 2011 [20]. According to the Institutional Financial Status and Expectations Report of the MHRT for the year 2022, In the first six months of 2022, 11082 White Code applications were processed in the White Code system and 6594 of those applications were considered legal aid eligible for action [21].

This study intends to evaluate the influencing factors of WPV events and their legal processing, as well as the frequency of WPV against HCWs.

## Methods

### Study design

This was an observational, descriptive, retrospective frequency study. This study was conducted at a tertiary medical faculty hospital with 1226 beds and approximately 3600 healthcare workers. Approximately one million people are admitted to our hospital every year as emergency patients, outpatients, and inpatients.

This study examined the WPV records of the Patient Rights and Employee Safety and Law departments of 135 healthcare professionals who made white-code applications between January 1, 2020, and March 31, 2022. The study population was not sampled. Sociodemographic characteristics (age, gender, and occupation), perpetrators of violence, types, places, times of violence, influencing factors (reasons), and legal process data were examined. Due to the absence of white code notifications, perpetrators'sociodemographic data could not be included in the study. The frequency of violence was calculated based on the number of health workers in the hospital annually. It was determined that there were approximately 3600 employees annually. The total frequency of violence was calculated by dividing the 135 violence records of the specified years by the cumulative total number of 10821 personnel. Factors affecting workplace violence were categorized as noncompliance with the procedure, communication, and dissatisfaction. Non-compliance with the service quality standards of the Ministry of Health was defined as "non-compliance with procedures". The causes of WPV arising from the communication between patients and their relatives and HCWs were defined as "communication". The causes of violence arising from the services provided to patients and their relatives are defined as "dissatisfaction".

### Verbal and physical violence

In the Turkish Criminal Code (TCC), insults and threats constitute verbal violent crimes; willful and willful killing constitute physical violent crimes. Insulting refers to attacking a person's honor, dignity, and dignity. An act or fact that may offend someone's dignity, honor, or dignity is attributed or cursed. A threat is anything that threatens the life, bodily integrity, or sexual well-being of another person or family. The penalty for defamation ranges from three

months to two years in prisons or judicial fines. Sentences ranging from six months to two years were imposed as a result of his threat. In TCC, willful injury and killing are defined as physical crimes. Intentionally causing pain to another person's body or deteriorating health or perception is punishable by imprisonment for one to three years. Intentionally killing someone is punishable in prison [22].

Verbal violence was classified as insulting-threatening or shouting-arguing in this study. Physical assault was defined as an attempt (throwing objects, attacking sharp objects, walking on) or physical contact (punching, kicking, twisting arms and hands, pushing, hitting the neck/shoulder).

## White code application

The white code application is a legal aid program offered by the MHRT and affiliated health institutions upon request of HCW in legal actions and proceedings undertaken within the scope of criminal law due to crimes committed against them during the delivery of health services by the MHRT according to legal regulations that entered into force in 2011. In addition, ongoing process transactions and judicial proceedings are recorded and followed in the white-code system by the MHRT. The purpose of the White Code is to facilitate the process of providing legal aid to HCWs. Additionally, the aim is to obtain statistical data that will be used to guide the fight against violence through root cause analyses [20,23]. The MHRT's official emergency code '1111' stands for workplace violence against healthcare providers in Turkey. HCWs who are exposed to WPV can call '1111' on their hospital phones. Alternatively, they can seek assistance from the official website of the MHRT White Code, or from the Patient Rights and Employee Safety unit of their hospital.Upon receiving a white code call, hospital security intervened at the scene. Regardless of complaints made by the health worker, the hospital administration, where the individual works, makes a criminal complaint to the Prosecutor's Office with all information and documents pertaining to the case. The prosecutor initiated a public judicial process. Additionally, according to the last legal regulation [24] that came into effect in 2022, university hospital lawyers are authorized to take legal action on behalf of victims' health workers or their legal representatives.

## Legal process

According to Turkish legal system, legal processes and definitions of violence against health workers reported with white code application.

Under investigation: The Prosecutor's Office is still investigating.

Investigation and non-prosecution: In cases where there is insufficient evidence to open a public action or where there is no chance of prosecution, the public prosecutor decides not to prosecute.

Judicial fine: In cases where there is no contrary provision in the law, the judicial fine is calculated by multiplying the total number of days determined as less than five days but not more than 730 days by the amount determined for one day, paid to the State Treasury by the convict.

Imprisonment: A prison sentence imposed by the court on a criminal is a punishment that restricts freedom. Term, life, and aggravated life imprisonment are types of imprisonment. Penitentiary institutions (prisons) are generally used for imprisonment purposes.

## Ethical considerations

This study was conducted in accordance with the principles of the Declaration of Helsinki. Approval was granted by the Non-Interventional Clinical Research Ethics Committee of the Dicle University Faculty of Medicine (June 09, 2022; number:183).

## Statistical analysis

Data analysis was conducted using the SPSS software (SPSS Statistics for Windows, version 24.0). Descriptive statistics are presented as percentages (%) and numbers (n), and continuous variables are expressed as mean ± standard deviation (min-max). chi-square ($\chi2$) test was used to analyze categorical variables, and Fisher's exact test was performed when more than 20% of the cells had frequencies less than 5. As a result of the analysis, $p<0.05$ was considered statistically significant.

## Results

The mean age of the 135 participants was 33.32±7.50 (min 23- max 60). WPV frequency was observed in a cumulative total of 10821 HCWs at 1.2% (n = 135). Any HCWs with more than one experience of violence were not found in the violence records. In terms of WPV type, 71.9%(n = 97) were verbal and 28.1%(n = 38) were physical. It was reported that 79.6% (n = 43) of women were exposed to verbal WPV, 20.4% (n = 11) to physical WPV, and 66.7% (n = 54) of men were exposed to verbal WPV, and 33.7% (n = 27) to physical WPV. In terms of exposure to WPV, doctors accounted for 62.3%, nurses for 20%, and medical secretaries for 7.4%. Most cases were observed in outpatient clinics (34.8%), followed by emergency departments (25.9%), during daylight hours (66.7%), and at a similar frequency (approximately 25.0%) during all seasons. All of the WPV events were perpetrated by the patients (54.8%) and their relatives (45.2%). Legal aid was provided to all white-code notifications. 37.1% were not prosecuted, 55.5% were under investigation, 4.4% were in accepting indictments, and 3.0% were punished by a judicial fine (Table 1). Among the WPV events, 85.2% (n = 115) targeted a single person and 14.8% (n = 20) targeted more than two individuals.

Based on the type of violence, there were no statistically significant differences between age, sex, occupation, place of occurrence, time, reason, and legal consequences ($p>0.05$) (Table 2). Among the main reasons for WPV against HCWs, noncompliance with procedures (49.6%), communication (27.4%), and dissatisfaction (23.1%) were identified. The most common reason for non-compliance with procedures was non-compliance with triage (25.7%) in the emergency department, request for priority examination in the outpatient clinic (17.1%), referral, visit, accompaniment, non-compliance with the payment rule in internal/surgical service and intensive care unit (37.4% internal services, 13.5% surgical services). Communication problems were most frequently caused by stressful patient management in the emergency room (14.2%), outpatient clinics (10.6%), surgical services (18.7%), or intensive care units (16.3%, respectively). Dissatisfaction in the emergency department was caused by long cues of services (14.3%) and HCWs' apathy (11.4%); in the outpatient clinic, it was caused by patient impatience, fussy (12.8%), and long cues of services (6.4%) (Table 3).

## Discussion

The healthcare field includes physicians, nurses, technicians, and others who are in direct contact with patients and visitors. In hospitals and healthcare facilities worldwide, patients and visitors are reported to have increasingly perpetrated verbal and physical violence against HCWs [2,5–8]. There are a number of adverse consequences associated with WPV for HCWs, including increased psychological stress, diminished job satisfaction, and decreased productivity [10,25,26]. Therefore, defining the frequency and affecting factors of WPV among healthcare professionals is essential before developing prevention policies and interventions.

A total of 4% of the active HCW population in European countries reported that they had been subjected to verbal or physical violence by visitors or patients [7]. In the present study, WPV frequency was observed in a cumulative total of 10821 healthcare workers at 1.2%). In

**Table 1. Sociodemographic, department, and judicial processes of workplace violence against Healthcare workers by each year.**

| | 2020 | 2021 | 2022***** | Total |
|---|---|---|---|---|
| | n(%) | n(%) | n(%) | n(%) |
| **Age Groups (year)** | | | | |
| 23–30 | 6(35.3) | 44(47.3) | 9(36.0) | 59(43.7) |
| 31–40 | 9(52.9) | 35(37.6) | 11(44.0) | 55(40.7) |
| >40 | 2(11.8) | 14(15.1) | 5(20.0) | 21(15.6) |
| **Sex** | | | | |
| Male | 11(64.7) | 55(59.1) | 15(60.0) | 81(60.0) |
| Female | 6(35.3) | 38(40.9) | 10(40.0) | 54(40.0) |
| **Occupation** | | | | |
| Physcian | 11(64.7) | 58(62.4) | 15(60.0) | 84(62.3) |
| Nurse | 4(23.5) | 19(20.4) | 4(16.0) | 27(20.0) |
| Medical Secretary | 1(5.9) | 8(8.6) | 1(4.0) | 10(7.4) |
| Technician | - | 3(3.2) | 3(12.0) | 6(4.4) |
| Others* | 1(5.9) | 5(5.4) | 2(8.0) | 8(5.9) |
| **Presence of Violence** | | | | |
| Yes | 17(0.5) | 93(2.5) | 25(0.7) | 135(1.2) |
| No | 3652(99.5) | 3566(97.5) | 3603(99.3) | 10821(98.8) |
| **Verbal Violence** | | | | |
| Insulting-Threatening | 4(23.5) | 19(20.4) | 3(12.0) | 26(19.3) |
| Shouting-Arguing | 13(76.5) | 74(79.6) | 22(88.0) | 109(80.7) |
| **Physicial Violence** | | | | |
| Attemp** | 2(50.0) | 11(42.3) | 3(60.0) | 16(42.1) |
| Physical Contact*** | 2(50.0) | 15(57.7) | 5(40.0) | 22(57.9) |
| **Perpetrator** | | | | |
| Patients | 9(52.9) | 50(53.8) | 15(60.0) | 74(54.8) |
| Visitors | 8(47.1) | 43(46.2) | 10(40.0) | 61(45.2) |
| **Department** | | | | |
| Emergency Department | 6(35.3) | 22(23.6) | 7(28.0) | 35(25.9) |
| Outpatient Clinic | 7(41.2) | 34(36.6) | 6(24.0) | 47(34.8) |
| Internal Ward / Intensive Care Unit | 1(5.9) | 11(11.8) | 4(16.0) | 16(11.9) |
| Surgical Ward / Intensive Care Unit | 3(17.6) | 26(28.0) | 8(32.0) | 37(27.4) |
| **Time (Hour)** | | | | |
| 08–16 | 8(47.0) | 68(73.1) | 14(56.0) | 90(66.7) |
| 17–24 | 7(41.2) | 19(20.4) | 9(36.0) | 35(25.9) |
| 00–08 | 2(11.8) | 6(6.5) | 2(8.0) | 10(7.4) |
| **Time (Season)** | | | | |
| Spring | 3(17.7) | 21(22.6) | 11(44.0) | 35(25.9) |
| Summer | 5(29.4) | 29(31.1) | - | 34(25.2) |
| Autumn | 5(29.4) | 25(26.9) | 1(4.0) | 31(23.0) |
| Winter | 4(23.5) | 18(19.4) | 13(52.0) | 35(25.9) |
| **Judicial Process** | | | | |
| Under Investigation | 8(47.0) | 53(57.0) | 14(56.0) | 75(55.5) |
| No- Investigation | - | 2(2.2) | - | 2(1.5) |
| No-Prosecuted | 6(35.3) | 31(33.3) | 11(44.0) | 48(35.6) |
| Accepting Indictment | 2(11.8) | 4(4.3) | - | 6(4.4) |
| Judicial Fine | 1(5.9) | 3(3.2) | - | 4(3.0) |

(*Continued*)

**Table 1.** (Continued)

|  | 2020 | 2021 | 2022***** | Total |
|---|---|---|---|---|
|  | n(%) | n(%) | n(%) | n(%) |
| Total**** | 17(12.6) | 93(68.9) | 25(18.5) | 135(100.0) |

*: Security (3), Cleaning staff (3), Support staff (2)

**: Throwing objects, attacking with sharp objects, walking on

***: Punching, kicking, twisting arms and hands, pushing, hitting the neck/shoulder

****:Row Percent (%)

*****:First 3 months.

terms of violence types, 71.9% were verbal and 28.1% were physical. This difference in the frequency of WPV can be attributed to the fact that it is often considered part of the HCW's job, leading to general underreporting in Turkey. Based on a national study by Pinar et al conducted with a survey method about WPV in Turkey, it was determined that 6.8% of health workers were exposed to physical violence, 43.2% to verbal violence, and 44.7% to both forms of violence [8]. As a result of the different study methods, direct comparisons with national studies may lead to misinterpretations. Pinar et al. investigated WPV against HCWs using a survey method. The present study focuses on the White Code System for WPV. It is possible that underreporting of WPV cases against healthcare workers contributes to the difference between survey studies and reporting mechanisms [11,12,27]. A total of 11082 applications for the White Code system were made in cases of WPV against healthcare professionals (over 1 million) in the first half of 2022, according to the report of the Turkish Ministry of Health [21]. The present study is more comparable with the report described above because we used the same definition of violence against HCWs and the same system of recording such violent events.

According to the literature, the most vulnerable HCWs in terms of exposure to WPV are those who provide direct contact with patients, their families, and visitors [11,12,28,29]. Cai et al investigated the occurrence of patient-initiated WPV based on HCW occupation. According to their findings, the vast majority of WPV cases involved doctors (72.6%) followed by nurses (14.3%) [29]. Abodunrin et al conducted a descriptive survey of the WPV. Researchers found that nurses account for 53.5% of WPV cases, followed by doctors with 21.5%) [30]. The WPV in Turkey was examined using a survey method in a national study. 72% of physicians and dentists, and 57.4% of nursing and midwifery professionals reported having experienced at least one form of WPV [8]. In terms of exposure to WPV, doctors accounted for 62.3%, nurses for 20%, and medical secretaries for 7.4% in the present study. In line with the literature, doctors and nurses in direct contact with patients were most likely to encounter WPV in this study.

Polat and Çırak conducted at a tertiary hospital, 345 cases of violence were examined based on white code reports. According to the study, the unit-based distribution of White Code notifications was 42.05% in the emergency department with the highest density, 26.66% within the clinics, specifically in surgical clinics, and 23.47% in the outpatient clinics [31]. Based on a study of 122 white code notifications over two years in a Turkish city, in terms of unit-based distribution, 31.1% of WPV cases occurred in emergency departments, and 25.4% in outpatient clinics [32]. In the present study, four clinical departments were used to categorize WPV cases. Most cases were seen in outpatient clinics (34.8%), followed by emergency departments (25.9%). In outpatient clinics and emergency departments, the high number of admissions, anxious nature of patients or their relatives, and overwhelming stress levels associated with waiting for diagnosis and treatment procedures may explain the prevalence of WPV.

**Table 2. In relation to the type of violence with the sociodemographic, departmental, and judicial processes of workplace violence against healthcare workers.**

| | Verbal n(%) | Physicial n(%) | p** |
|---|---|---|---|
| **Age Group (Year)** | | | |
| 23–30 | 38(39.2) | 21(55.3) | >0.05 |
| 31–40 | 42(43.3) | 13(34.2) | |
| >40 | 17(17.5) | 4(10.5) | |
| **Sex** | | | |
| Male | 54(55.7) | 27(71.1) | >0.05 |
| Female | 43(44.3) | 11(28.9) | |
| **Occupation** | | | |
| Physician | 59(60.8) | 25(65.8) | >0.05 |
| Non- Physician | 38(39.2) | 13(34.2) | |
| **Department** | | | |
| Emergency Department | 24(24.7) | 11(28.9) | >0.05 |
| Outpatient Clinic | 37(38.2) | 10(26.3) | |
| Internal Ward / Intensive Care Unit | 9(9.3) | 7(18.4) | |
| Surgical Ward / Intensive Care Unit | 27(27.8) | 10(26.3) | |
| **Time (Year)** | | | |
| 2020 | 13(13.4) | 4(10.5) | >0.05 |
| 2021 | 67(69.1) | 26(68.4) | |
| 2022 (First 3 Month) | 17(17.5) | 8(21.1) | |
| **Time (Hour)** | | | |
| 08–16 | 69(71.1) | 21(55.3) | >0.05 |
| 17–24 | 22(22.7) | 13(34.2) | |
| 00–08 | 6(6.2) | 4(10.5) | |
| **Time (Season)** | | | |
| Spring | 28(28.9) | 7(18.4) | >0.05 |
| Summer | 21(21.6) | 13(34.2) | |
| Autumn | 25(25.8) | 6(15.8) | |
| Winter | 23(23.7) | 12(31.6) | |
| **Causes of Violence** | | | |
| Non-compliance with procedures | 46(47.5) | 21(55.3) | >0.05 |
| Communication | 27(27.8) | 10(26.3) | |
| Dissatisfaction | 24(24.7) | 7(18.4) | |
| **Judicial Process** | | | |
| Under Investigation | 52(53.6) | 23(60.5) | >0.05 |
| No- Investigation | 2(2.1) | - | |
| No-Prosecuted | 35(36.1) | 13(34.2) | |
| Accepting Indictment | 4(4.1) | 2(5.3) | |
| Judicial Fine | 4(4.1) | - | |
| **Total*** | **97(71.9)** | **38(28.1)** | |

*: Row %

**: Chi-square test/Fisher's exact test, non-significantly.

WPV has been reported to originate from patients and their families in many previous studies [28–32]. Consistent with literature, the present study found that patients and their families were the most likely perpetrators of WPV against HCWs (54.8% and 45.2%, respectively).

**Table 3. The causes of the violence that occurred in the departments.**

| Cause of Violence | Emergency Department n(%) | Outpatient Clinic n(%) | Internal Ward / Intensive Care Unit n(%) | Surgical Ward / Intensive Care Unit n(%) | Total n(%) |
|---|---|---|---|---|---|
| **Non-compliance with procedures** | **12(34.4)** | **27(57.5)** | **12(75.0)** | **16(43.2)** | **67(49.6)** |
| Non-compliance with triage | 9(25.7) | 1(2.1) | 2(12.5) | 4(10.8) | 16(11.9) |
| Request for priority examination | 1(2.9) | 8(17.1) | - | 3(8.1) | 12(8.9) |
| Referral, visit, accompaniment, non-compliance with the payment rule | 1(2.9) | 6(12.8) | 6(37.4) | 5(13.5) | 18(13.3) |
| Requests for services other than those recommended by the physician | 1(2.9) | 6(12.8) | 1(6.3) | 2(5.4) | 10(7.4) |
| Request a medical examination without an appointment | - | 5(10.6) | 2(12.5) | 1(2.7) | 8(5.9) |
| Non-compliance with privacy | - | 1(2.1) | 1(6.3) | 1(2.7) | 3(2.2) |
| **Communication** | **13(37.0)** | **9(19.1)** | **3(18.7)** | **12(32.5)** | **37(27.4)** |
| Death-related issues | 3(8.6) | 1(2.1) | - | 3(8.1) | 7(5.2) |
| Stressful patient management | 5(14.2) | 5(10.6) | 3(18.7) | 6(16.3) | 19(14.1) |
| Diagnosis and treatment refusal | 5(14.2) | 3(6.4) | - | 2(5.4) | 10(7.4) |
| Holding the healthcare workers responsible for the patient's treatment process | - | - | - | 1(2.7) | 1(0.7) |
| **Dissatisfaction** | **19(28.6)** | **11(23.4)** | **1(6.3)** | **9(24.3)** | **31(23.0)** |
| Long cue of service | 5(14.3) | 3(6.4) | - | 1(2.7) | 9(6.7) |
| Patient impatience (fussy) | 1(2.9) | 6(12.8) | 1(6.3) | 2(5.4) | 10(7.4) |
| Healthcare worker apathy | 4(11.4) | 1(2.1) | - | 3(8.1) | 8(5.9) |
| Quality of non-medical services | - | 1(2.1) | - | 3(8.1) | 4(3.0) |
| **Total*** | **35(25.9)** | **47(34.8)** | **16(11.9)** | **37(27.4)** | **135 (100.0)** |

*:Row %.

In a systematic review of WPV against HCWs, Keser et al stated that waiting times were an obvious reason in five of the 10 studies examining the causes of WPV, while excessive patient and family demands, low education levels, and non-compliance with rules were cited as contributing factors [15]. Among the main reasons for WPV against HCWs, noncompliance with procedures (49.6%), communication (27.4%), and dissatisfaction (23.1%) were identified in the present study. The most common reason for non-compliance with procedures was non-compliance with triage (25.7%) in the emergency department, request for priority examination in the outpatient clinic (17.1%), referral, visit, accompaniment, non-compliance with the payment rule in internal/surgical service and intensive care unit (37.4% internal services, 13.5% surgical services). Communication problems were most frequently caused by stressful patient management in the emergency room (14.2%), outpatient clinics (10.6%), surgical services (18.7%), or intensive care units (16.3%, respectively). Dissatisfaction in the emergency department was caused by long cues of services (14.3%) and HCWs' apathy (11.4%); in the outpatient clinic, it was caused by patient impatience, fussy (12.8%), and long cues of services (6.4%). The problem of WPV in hospitals and healthcare facilities is complex, heterogeneous, and multifactorial; however, it can also be prevented. The primary causes of WPV against HCWs in this study should be useful in reducing or eliminating this risk.

According to the MHRT report, 6,594 WPV cases (out of 11082 White Code applications) were evaluated within the scope of legal aid in the first half of 2022 [21]. Among 122 white code notifications in a Turkish city, 44.2% received legal aid [32]. Among white code

notifications for which legal aid was provided, 44.4% were not prosecuted, 38.9% were under investigation, and 16.7% were in the process of preparing or accepting an indictment according to research data. Legal aid was provided to all white-code notifications in the present study. However, 37.1% were not prosecuted, 55.5% were under investigation, 4.4% were in accepting indictments, and 3.0% were punished by a judicial fine. Legally, hospital management must file a criminal complaint with the Prosecutor's Office on behalf of the individual, regardless of any white-code applications or compliance with the HCW. Owing to a lack of information, healthcare workers may provide an incorrect white code application. Therefore, hospital administrations want to reduce bureaucratic procedures in unsuccessful judicial processes to prevent the initiation of erroneous legal aid procedures. As a result, the HCW's statement was applied again after the white code application. The Prosecutor's Office may not be notified when the HCW understands that the individual's application does not comply with the informed consent of the individual after being provided with white code notification. The high frequency of legal aid in this study may be due to the provision of legal aid for all white-code notifications. However, the length of the proceedings and the perceived lack of protection from legal regulations may have prompted the HCWS to underreport WPV events.

## Strengths and limitations

In the present study, which examined white code notifications to determine the frequency of WPV and preventable factors, may contribute to studies on WPV prevention. By evaluating judicial proceedings of HCWs reporting violence suffered, we examined whether the violence had a deterrent response in the judiciary.

Study limitations include the fact that the study was conducted at only one center. As HCWs chose whether to use white-code applications to report WPV cases, the results cannot be generalized and should be interpreted carefully. Only notified white-code records were included in this study. Incomplete information and documents were presented in the legal processes, and records of perpetrators were missing. The WPV phenomenon within health organizations should be explored more comprehensively in future research.

## Conclusion

The frequency of WPV reported in this study was lower than that reported in the literature. The majority of WPV incidents were directed toward HCWs at outpatient clinics and emergency services. As a result of many preventable factors, including a lack of objective patient expectations, problems with complying with hospital rules, challenging patient management, communication, excessive workload, and lengthy legal processes, WPV events were observed. The legal process took a long time to complete. Most of them did not lead to prosecution and very few resulted in punishment. The service areas should be evaluated to determine whether they meet the expectations. The potential contribution of health personnel communication to violence can be addressed through training. Legal arrangements can be made regarding deterrent penalties and shortened legal processes. In order to prevent workplace violence, health personnel can be encouraged to report it.

## Author Contributions

**Conceptualization:** Hıdır Sari, İsmail Yildiz, Senem Çağla Baloğlu, Mehmet Özel, Ronay Tekalp.

**Data curation:** Hıdır Sari, İsmail Yildiz, Senem Çağla Baloğlu, Mehmet Özel, Ronay Tekalp.

**Formal analysis:** Hıdır Sari, İsmail Yildiz, Senem Çağla Baloğlu, Mehmet Özel, Ronay Tekalp.

**Investigation:** Hıdır Sari, İsmail Yildiz, Senem Çağla Baloğlu, Ronay Tekalp.

**Methodology:** Hıdır Sari, İsmail Yildiz, Mehmet Özel, Ronay Tekalp.

**Writing – original draft:** Hıdır Sari.

**Writing – review & editing:** Hıdır Sari, İsmail Yildiz, Senem Çağla Baloğlu, Mehmet Özel, Ronay Tekalp.

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
