## [Decision Letter · Decision Letter 0]

27 Mar 2023

PONE-D-23-03576The frequency of workplace violence against healthcare workers and affecting factorsPLOS ONE

Dear Dr. Sari,

Thank you for submitting your manuscript to PLOS ONE. After careful consideration, we feel that it has merit but does not fully meet PLOS ONE’s publication criteria as it currently stands. Therefore, we invite you to submit a revised version of the manuscript that addresses the points raised during the review process.

 Please submit your revised manuscript by May 11 2023 11:59PM. If you will need more time than this to complete your revisions, please reply to this message or contact the journal office at plosone@plos.org. Please include the following items when submitting your revised manuscript:A rebuttal letter that responds to each point raised by the academic editor and reviewer(s). You should upload this letter as a separate file labeled 'Response to Reviewers'.A marked-up copy of your manuscript that highlights changes made to the original version. You should upload this as a separate file labeled 'Revised Manuscript with Track Changes'.An unmarked version of your revised paper without tracked changes. You should upload this as a separate file labeled 'Manuscript'.

We look forward to receiving your revised manuscript.

Kind regards,

Nadeeka Kumudini Chandraratne

Academic Editor

PLOS ONE

Journal Requirements:

Additional Editor Comments:

Dear Author,

The reviewers have gone through your submission and requested major revisions in your work. In addition, we would recommend you to attend to the following important aspects noted by us.

1. The methodology should be expanded. It does not give the essence of what you have actually conducted. (eg: Were the 135 participants questioned or was it only the records that were used?)

2. The funding sources should be declared. If no funding received, please indicate that.

3. The recommendations seems to be not arising from your findings. Please give recommendations based on your original work.

Reviewers' comments:

Reviewer's Responses to Questions

**Comments to the Author**

1. Is the manuscript technically sound, and do the data support the conclusions?

Reviewer #1: Partly

Reviewer #2: Yes

2. Has the statistical analysis been performed appropriately and rigorously? 

Reviewer #1: N/A

Reviewer #2: Yes

3. Have the authors made all data underlying the findings in their manuscript fully available?

Reviewer #1: Yes

Reviewer #2: Yes

4. Is the manuscript presented in an intelligible fashion and written in standard English?

Reviewer #1: Yes

Reviewer #2: Yes

5. Review Comments to the Author

Reviewer #1: Thanks for the opportunity to review the manuscript titled “The frequency of workplace violence against healthcare workers and affecting factors”. The manuscript addresses an important and relevant issue regarding e the influencing factors of workplace violence (WPV) events and their legal processing, as well as the frequency of WPV against HCWs. The authors performed an observational, descriptive, retrospective frequency study. The objectives are stated clearly, but methods are insufficient and unclear. The small sample size (n. 135 healthcare workers) does not allow to draw significant conclusions. Therefore, the aforementioned limitation does not allow to attribute significance to the results of the study. The authors did not assess the way to minimize such limitation of the study.

Reviewer #2: Thank you for the opportunity to review this article. The topic is certainly interesting and relevant. Some minor changes are needed before the article can be published.

Much of the article is well written. I am not sure why the authors have included Table 1 as it does not provide relevant information. It is also not clear if the statistics in Table 1 are related to the hospital that they are researching or to hospitals in general. In the introduction, they refer twice to workplace violence happening to ‘important people’ – I am not sure who they mean by this, and it would be good to clarify. In the ‘Legal Process’ section, it would be good to have a few sentences at the start explaining the section. In the abstract and in the results section after you mention 1.2%, you have a bracket ‘)’ that needs to be removed.

The authors stated that they organised the data within the categories of ‘non compliance with procedure, communication and dissatisfaction’. It would be good to clarify how and why they came up with these 3 headings as it seems that they were pre-selected by the authors.

In terms of the findings, more clarification is needed as to why 135 healthcare workers were included in the study, but then 10821 healthcare workers were reported as the cumulative total. Is it that there were 10821 incidences? Please clarify this. If it is only 135 health care workers, then there are likely to be healthcare workers with multiple experiences of violence and this really needs to be discussed and acknowledged.

In the discussion section, in the paragraph where you begin with ‘In a tertiary hospital, 345 cases of violence were examined based on white code reports…’, are you talking about the study’s hospital or another hospital? In the discussion it would also be good to talk about the influence of culture on workplace violence as your study is specific to Turkey. Is there an influence of cultural expectations that are resulting in these findings?

You also state that the study will make ‘important contributions to the development of prevention policies by evaluating judicial proceedings of reported violence suffered’- I don’t believe this argument can be made so strongly. In your study you have identified the causes of violence and some of the consequences, which could provide insight when educating current and future health professionals. I think you need to clearly identify how your findings can affect policies and it would be good to have more recommendations emerging from your findings.

With the above changes, I believe your article will be stronger and ready for publication.

Thank you.

6. PLOS authors have the option to publish the peer review history of their article (what does this mean?). If published, this will include your full peer review and any attached files.

Reviewer #1: No

Reviewer #2: No

---

## [Author Response · Author response to Decision Letter 0]

17 Apr 2023

03-April-2023

Dear Editor,

Respond to the reviewers’ comments for the manuscript ID PONE-D-23-03576 entitled "The frequency of workplace violence against healthcare workers and affecting factors"

First of all, we would like to thank you, your editorial staff, and the expert reviewers for your very valuable comments, feedback, and suggestions. During our review of your changes and suggestions, we paid close attention to each one. Below are our responses to the changes and suggestions. It is our hope that the changes we have made to the redressed version of our manuscript will be acceptable for publication in your respected journal after the revisions we have made.

Sincerely yours, 

Hıdır SARI, MD.

Response to Additional Editor Comments:

1. The methodology should be expanded. It does not give the essence of what you have actually conducted. (eg: Were the 135 participants questioned or was it only the records that were used?)

Response 1: By extending the methodology, relevant changes to the paper are marked in yellow

2. The funding sources should be declared. If no funding received, please indicate that.

Response 2: “The authors received no specific funding for this work.” sentence was added in Declaration section of manuscript.

3. The recommendations seems to be not arising from your findings. Please give recommendations based on your original work.

Response 3: Recommendations in line with the study findings were made and added to the conclusion section.

Response to Reviewers' comments:

Reviewer's Responses to Questions

Comments to the Author

1. Is the manuscript technically sound, and do the data support the conclusions?

Reviewer #1: Partly

Reviewer #2: Yes

2. Has the statistical analysis been performed appropriately and rigorously?

Reviewer #1: N/A

Reviewer #2: Yes

3. Have the authors made all data underlying the findings in their manuscript fully available?

Reviewer #1: Yes

Reviewer #2: Yes

4. Is the manuscript presented in an intelligible fashion and written in standard English?

Reviewer #1: Yes

Reviewer #2: Yes

5. Review Comments to the Author

Reviewer #1: Thanks for the opportunity to review the manuscript titled “The frequency of workplace violence against healthcare workers and affecting factors”. The manuscript addresses an important and relevant issue regarding e the influencing factors of workplace violence (WPV) events and their legal processing, as well as the frequency of WPV against HCWs. The authors performed an observational, descriptive, retrospective frequency study. The objectives are stated clearly, but methods are insufficient and unclear. The small sample size (n. 135 healthcare workers) does not allow to draw significant conclusions. Therefore, the aforementioned limitation does not allow to attribute significance to the results of the study. The authors did not assess the way to minimize such limitation of the study.

Response to Reviewer #1: The study was conducted retrospectively through hospital records, so all violence records of the Patient Rights and Employee Safety and Law departments were analyzed. There were only 135 reports of violence among healthcare workers. This was described in the method section

Reviewer #2: Thank you for the opportunity to review this article. The topic is certainly interesting and relevant. Some minor changes are needed before the article can be published.

Much of the article is well written. I am not sure why the authors have included Table 1 as it does not provide relevant information. It is also not clear if the statistics in Table 1 are related to the hospital that they are researching or to hospitals in general. 

Response to Reviewer #2: Table 1 was removed from article.

In the introduction, they refer twice to workplace violence happening to ‘important people’ – I am not sure who they mean by this, and it would be good to clarify. 

Response to Reviewer #2: Occupational accident and violence definition was used in the introduction of the manuscript according to a report published by the International Labor Organization (ILO). The relevant definition sentence has been reorganized in the introduction.

In the ‘Legal Process’ section, it would be good to have a few sentences at the start explaining the section. 

Response to Reviewer #2: ‘’According to Turkish legal system, legal processes and definitions of violence against health workers reported with white code application.’’ sentence was added.

In the abstract and in the results section after you mention 1.2%, you have a bracket ‘)’ that needs to be removed.

Response to Reviewer #2: Have been corrected.

The authors stated that they organised the data within the categories of ‘non compliance with procedure, communication and dissatisfaction’. It would be good to clarify how and why they came up with these 3 headings as it seems that they were pre-selected by the authors.

Response to Reviewer #2: These definitions were added and described in the method section.

In terms of the findings, more clarification is needed as to why 135 healthcare workers were included in the study, but then 10821 healthcare workers were reported as the cumulative total. Is it that there were 10821 incidences? Please clarify this. 

Response to Reviewer #2: This clarification was added and described in the method section. 

If it is only 135 health care workers, then there are likely to be healthcare workers with multiple experiences of violence and this really needs to be discussed and acknowledged.

Response to Reviewer #2: ‘’Any HCWs with more than one experience of violence were not found in the violence records’’. 

In the discussion section, in the paragraph where you begin with ‘In a tertiary hospital, 345 cases of violence were examined based on white code reports…’, are you talking about the study’s hospital or another hospital? 

Response to Reviewer #2:’’ Polat and Çırak conducted at a tertiary hospital , 345 cases of violence were examined based on white code reports. ‘’ This sentences added in the discussion section.

In the discussion it would also be good to talk about the influence of culture on workplace violence as your study is specific to Turkey. Is there an influence of cultural expectations that are resulting in these findings?

Response to Reviewer #2: There may be cultural reasons behind the reasons for violence reported by healthcare professionals, but in this study, when the records of victims of violence were examined, no cultural records were found among the reasons for violence. This interesting reason could be a different research topic.

You also state that the study will make ‘important contributions to the development of prevention policies by evaluating judicial proceedings of reported violence suffered’- I don’t believe this argument can be made so strongly.

Response to Reviewer #2: Edited in the strengths section according to the reviewer's comment. 

In your study you have identified the causes of violence and some of the consequences, which could provide insight when educating current and future health professionals. I think you need to clearly identify how your findings can affect policies and it would be good to have more recommendations emerging from your findings.

Response to Reviewer #2: Have been explained and added in the conclusion section.

With the above changes, I believe your article will be stronger and ready for publication.

Thank you.

6. PLOS authors have the option to publish the peer review history of their article (what does this mean?). If published, this will include your full peer review and any attached files.

Do you want your identity to be public for this peer review? For information about this choice, including consent withdrawal, please see our Privacy Policy.

Reviewer #1: No

Reviewer #2: No

Data Availability Statement

Due to research ethics committee restrictions, we are unable to make the data publicly available due to restrictions in the approved protocol. Although no personally identifiable data were collected and the participants' consent form did not inform the participants that their data would be shared outside the research team, even if their identity information was removed. For more information, please contact Dicle University Ethics Committee at kuruletikdiyar@gmail.com.

---

## [Decision Letter · Decision Letter 1]

18 Jul 2023

The frequency of workplace violence against healthcare workers and affecting factors

PONE-D-23-03576R1

Dear Dr. SARI,

We’re pleased to inform you that your manuscript has been judged scientifically suitable for publication and will be formally accepted for publication once it meets all outstanding technical requirements.

Kind regards,

Andrea Cioffi

Academic Editor

PLOS ONE

Additional Editor Comments (optional):

Reviewers' comments:

Reviewer's Responses to Questions

**Comments to the Author**

1. If the authors have adequately addressed your comments raised in a previous round of review and you feel that this manuscript is now acceptable for publication, you may indicate that here to bypass the “Comments to the Author” section, enter your conflict of interest statement in the “Confidential to Editor” section, and submit your "Accept" recommendation.

Reviewer #1: All comments have been addressed

Reviewer #2: All comments have been addressed

2. Is the manuscript technically sound, and do the data support the conclusions?

Reviewer #1: Yes

Reviewer #2: Yes

3. Has the statistical analysis been performed appropriately and rigorously? 

Reviewer #1: Yes

Reviewer #2: Yes

4. Have the authors made all data underlying the findings in their manuscript fully available?

Reviewer #1: Yes

Reviewer #2: Yes

5. Is the manuscript presented in an intelligible fashion and written in standard English?

Reviewer #1: Yes

Reviewer #2: Yes

6. Review Comments to the Author

Reviewer #1: Author answered all my questions. I think the manuscript is ready for publication. Policymakers and healthcare workers could benefit from reading this manuscript.

Reviewer #2: Happy to accept. They addressed my comments. It is a good article. I wish the authors all the best in their future research.

7. PLOS authors have the option to publish the peer review history of their article (what does this mean?). If published, this will include your full peer review and any attached files.

Reviewer #1: No

Reviewer #2: No

---

## [Editor Report · Acceptance letter]

21 Jul 2023

PONE-D-23-03576R1 

The frequency of workplace violence against healthcare workers and affecting factors 

Dear Dr. Sari:

I'm pleased to inform you that your manuscript has been deemed suitable for publication in PLOS ONE. Congratulations! Your manuscript is now with our production department. 

Kind regards, 

on behalf of

Dr. Andrea Cioffi 

Academic Editor

PLOS ONE